# An Experimental and Numerical Investigation into the Durability of Fibre/Polymer Composites with Synthetic and Natural Fibres

**DOI:** 10.3390/polym14102024

**Published:** 2022-05-16

**Authors:** Abdalrahman Alajmi, Rajab Abousnina, Abdullah Shalwan, Sultan Alajmi, Golnaz Alipour, Tafsirojjaman Tafsirojjaman, Geoffrey Will

**Affiliations:** 1School of Chemistry, Physics and Mechanical Engineering, Queensland University of Technology, Brisbane, QLD 4000, Australia; mr.gz@hotmail.com (A.A.); g.will@qut.edu.au (G.W.); 2Department of Power and Desalination Plants, Ministry of Electricity and Water, Kuwait City 12010, Kuwait; ajmi.q8@hotmail.com; 3School of Engineering, Faculty of Science and Engineering, Macquarie University, Sydney, NSW 2109, Australia; golnaz.alipour@mq.edu.au; 4Department of Manufacturing Engineering Technology, Public Authority for Applied Education and Training, Kuwait City 13092, Kuwait; ama.alajmi1@paaet.edu.kw; 5Centre for Future Materials (CFM), School of Civil Engineering and Surveying, University of Southern Queensland, Toowoomba, QLD 4350, Australia; tafsirojjaman@usq.edu.au

**Keywords:** composites, biodegradation, hydrothermal ageing, durability, natural fibres, synthetic fibres

## Abstract

Progress in engineering research has shifted the interest from traditional monolithic materials to modern materials such as fibre reinforced composites (FRC). This paradigm shift can be attributed to the unique mechanical characteristics of FRCs such as high strength to weight ratio, good flexural strength, and fracture toughness. At present, synthetic composites dominate the automotive, aerospace, sporting, and construction industries despite serious drawbacks such as costly raw materials, high manufacturing costs, non-recyclability, toxicity, and non-biodegradability. To address these issues, naturally occurring plant fibres (such as jute, hemp, sisal) are being increasingly researched as potential reinforcements for biodegradable or non-biodegradable polymer matrices to produce environmentally friendly composites. In this study, sisal fibres were selected owing to their low production costs, sustainability, recyclability, and biodegradability. The hydrothermal ageing and mechanical characteristics of sisal fibre-reinforced epoxy (SFRE) composites were determined and compared with glass fibre-reinforced epoxy (GFRE) synthetic composites. Moreover, a first-of-its-kind numerical model have been developed to study the hydrothermal ageing and mechanical characteristics of SFRE, along with GFRE, using ANSYS software. Moreover, microstructural analysis of flexural tested GFRE and SFRE samples were carried out to identify the microstructural properties of the composites. Both experimental and numerical results exhibited an influence of short- or long-term hydrothermal treatment on the flexural properties of glass and sisal fibre-based composites. In the case of GFRE, the moisture uptake and fibre-matrix de-bonding existed, but it is less severe as compared to the SFRE composites. It was found that the dosage of sisal fibres largely determines the ultimate mechanical performance of the composite. Nonetheless, the experimental and numerical flexural strengths of SFRE were comparable to GFRE composites. This exhibited that the SFRE composites possess the potentiality as a sustainable material for advanced applications.

## 1. Introduction

Research and engineering advancements have redirected emphasis away from traditional monolithic materials and focuses towards novel materials such as fibre reinforced composites (FRC). The distinctive mechanical properties of FRC, including high tensile strength, strong flexural modulus, and impact resistance, have caused this massive change. A low-strength polymer matrix is reinforced with greater strength fibres ranging from glass to carbon. Although completely synthetic composites dominate the automobiles, aircraft, sporting goods, and infrastructure sectors [1], they have significant disadvantages such as high input costs, high cost of production, non-recyclability, toxicity, and non-biodegradable [2,3].

On the other hand, polymer matrices are widely used due to their useful characteristics which include: lightweight, attractive optical properties, ease of shaping, high material stability and good thermal and electrical insulation. However, they also have limitations such as low modulus and low service temperatures. When fabricating an FRC, the polymer matrix must be carefully selected for its intended application since the matrix binds the individual fibres together into an integral unit, enhancing the load-carrying capability of the composite, and protecting the fibres from mechanical abrasion and environmental degradation. These polymers are amorphous or semi-crystalline in nature and consist of linear or branched chains bonded by strong intramolecular but weak inter-molecular forces; this means they can be moulded and remoulded easily by varying the heat and pressure. However, their applications in FRCs are limited due to their low service temperatures [4,5]. Other thermoplastics such as low-density polyethylene, high-density polyethylene, polypropylene, polystyrene, polyvinyl chloride, a blend of different polymers, and recycled thermoplastics are used as matrix materials [5,6]. These polymers are stiff and cannot be remoulded since their network structure has intramolecular covalent bonds [7]. Therefore they cannot be reshaped and decompose when heated to elevated temperatures. Common examples include unsaturated polyester resins [8], phenolic epoxy, epoxide resins, vinyl esters, silicone, and polyimides [9]. Furthermore, in an FRC, reinforced fibres are the largest proportion, and act as the major load-carrying portion of any composite. Fibre is commonly an elongated discrete thread or a bundle of continuous filaments. When selecting the type of fibre, factors such as the material volume fraction, length, and alignment of the fibre must be considered since they impart several characteristics to the final composite [3]. An FRC can be tailor-made using Orthotropic (fibres are arranged in rectangular forms) or Quasi-isotropic (fibres are randomly oriented) for any specific application. Fibres can be broadly classified as synthetic or natural; common synthetic fibres include glass, carbon, and aramid fibres. While natural fibres such as silk, wool, hemp, flax, jute, kenaf, sisal, and banana can be reinforced into FRCs [10].

Synthetic fibres can be made from inorganic and organic raw materials. Glass fibres, carbon fibres, aramid fibres, and boron fibres are the types of synthetic fibres that are widely used as reinforcing material [10]. Inorganic fibres such as glass and carbon are commonly synthesised by heating the raw materials, while organic fibres are synthesised through polymerization methods i.e., solution or bulk polymerization [11]. Once synthesised, the fibres are further treated to form different varieties. The glass then melted in a furnace is directly formed into fibres such as continuous filaments, staple fibres, and chopped fibres [12]. FRC containing synthetic fibres has a high strength to weight ratio, good flexural strength, and fracture toughness. Besides this, they also have limitations such as non-recyclability, toxicity, and non-biodegradability [2,3]. Due to global warming, some countries have implemented environmental legislation that restricts the use of non-recyclable and non-disposable composite materials [2], with the result that naturally occurring fibres such as jute [13,14,15,16], hemp [17], sisal [16,18], etc. are being widely researched as reinforcements for making environment-friendly composite materials. These composites possess properties such as sustainable raw materials, recyclability, and biodegradation.

Natural fibres, also known as lignocellulosic materials, have long been used as reinforcing materials due to their numerous benefits, including high strength, low density, biodegradability ease of handling, renewable resource, electric, thermal, and acoustic insulation, aesthetic aspects, and non-toxicity. In recent years, there has been an increasing interest in using natural fibres as a reinforcing agent in composites to build low-cost construction materials [19]. Their recent history can be traced back to 1908 when they were first used to reinforce phenolic resins and other materials such as urea and melamine etc. [20]. Plant-based natural fibres are in great demand in the building construction and automotive sectors due to their low density and less cost. Due to these benefits and applications of plant fibres, this study has focused on the utilization of plant fibres as reinforcing fibres [21]. Moreover, the health risks of asbestos fibre, high energy consumption, non-biodegradable nature and high cost of synthetic fibres (plastic, glass, Kevlar, carbon) limits its use in fibre reinforced composites [22]. Natural fibres are divided into three main categories depending upon their plant origin, seed fibres (cotton, oil palm, coir, etc.), stem fibres (examples include hemp, flax, jute, kenaf etc.), and leaf fibres (sisal, pineapple, banana, bamboo, bagasse, etc.) [10,14]. Plant fibres are also called lignocellulose fibres due to their cellulosic chemical composition [10]. Their building block is cellulose dispersed as microfibrils in an amorphous medium that is referred to as lignin. Depending upon the type of fibre, the percentage of cellulose can vary between 60 to 75% while lignin ranges from 5 to 15% by weight. Other components such as hemicellulose, pectin, waxes, and moisture are also found in plant fibres in different proportions [23,24]. The chemical properties of natural fibres are largely determined by their age, source and weather conditions and other factors. In the polymer matrix, lignocellulosic fibres such as sisal, banana, palmyra, coir, pineapple, jute, hemp and kenaf have been widely employed [25]. Among them, sisal fibres have been widely used due to its excellent mechanical properties i.e., strength, stiffness modulus which are listed in Table 1. Ropes, carpets, fishnets, cables, bag fabrics, cushioning, and carpets are all made from sisal fibre [26]. Moreover, internal engine covers of the engine, door handles, hat racks, sun view mirrors, seatbacks, package trays and external or underfloor panelling can all benefit from sisal fibre-based composites, according to many studies. They’ve been employed in the aerospace and aircraft industries for interior panels. They’ve been employed in the aerospace and aircraft sectors for interior panels. Sisal fibre comprises of hemicellulose, lignin, cellulose and moisture (Table 2). Since it is extracted from lignocelluloses, it is hydrophilic and includes strongly polarised -OH groups. Impurities such as wax and natural oils are a serious issue with sisal fibre [27]. Poor adhesion between hydrophilic fibre (polar) and hydrophobic matrix (non-polar) is a major drawback using sisal fibre as a reinforcing material [26]. The micromechanical performance of composites is influenced by the fibre/matrix contact [22]. So, the surface of sisal fibre is modified by alkali treatment in order to overcome some of the drawbacks associated with the application of lignocellulosic fibres and to enhance the interfacial contact between fibre and polymer matrix [28,29,30]. Sisal fibres are widely used due to their easy cultivation and short renewal times. Each year, an estimated 5 million tons of sisal fibres are produced around the globe, most of which comes from Brazil and Tanzania. They are extracted from a plant called Agave Sisalana, which is native to the West Indies, Africa, and sub-tropical regions such as South and North America. A sisal leaf is commonly a source of ribbon fibres, mechanical fibres, and xylem fibres [31]. The mechanical fibres are the most useful for commercial-scale applications since they mainly come from the leaf’s periphery. Ribbon fibres are longer than other types. In addition to that, the sisal microstructure shows clear signs of degradation indicating that part of the material that composes the fibre cell wall degenerated [32]. However, the composite degradation is out of the scope of this study.

Each sisal plant leaf contains 1000 to 1200 bundles of fibres. These bundles consist of 4% fibres, 8% dry matter, 0.75% cuticle, and around 88% of water. The fibres are extracted by retting and scraping or by mechanical extraction. Once the fibres are extracted, they are washed with clean water to remove the leaf juices, adhesive solids, chlorophyll, and so on [31,35]. Sisal fibres have good insulation properties, biodegradation, and recyclability so they are an inexpensive option (available at ~US$; 1 per kg) that cost around 1/9 of glass fibres. However, the composite degradation is out of the scope in this study. Sisal fibres in a high alkaline environment can discolor and lose a substantial proportion of their mechanical strength, whereas, at higher temperatures and in an alkaline environment the fibres can rapidly decompose. To eradicate these issues, it has been suggested that the composite be treated with carbonate [28,31]. The mechanical characteristics of natural fibre composites (NFCs) depend on the type of fibre used in a particular thermoplastic resin. NFCs are generally very sensitive to environmental variables such as the moisture content in the surrounding air. Their strength is generally lower than their synthetic counterparts since they are less brittle and not as stiff. It was also found when jute fibre was used as a reinforcement, its tensile strength depended directly on the cross-sectional area of the fibre and the composite decomposed when exposed to ambient environments, whereas with sisal fibre, the overall tensile strength, impact strength, and flexural strength improved [36]. In common applications, FRCs are exposed to environments ranging from variations in the surrounding temperature to changes in the level of humidity. These composites were also exposed to very hot and wet conditions. In these circumstances, the mechanisms and effects of hydrothermal ageing on common composites must be understood since they enable scientists to predict the durability and service life of a specific composite. Several researchers [37,38] studied the effect of hydrothermal ageing on the physical and mechanical properties of FRCs and found that the penetration of moisture into the microstructures is a major limitation for their industrial usage.

This phenomenon is quite prevalent for composites reinforced with natural fibres, in fact, according to Hu, Sun [39], the mechanical properties of polylactide reinforced with short jute fibres are significantly affected by long or short-term hydrothermal ageing. An analysis by electron microscope revealed that the fracture and debonding of fibres under these circumstances are the underlying damage mechanisms. Assarar, Scida [40] fabricated composites consisting of a polypropylene matrix reinforced with short hemp fibres, exposed them to various environmental conditions and studied the environmental impact on their mechanical characteristics. They observed a decrease in the tensile strength and Young’s modulus of the aged composite samples. NFCs were also subjected to testing by immersion in distilled water for a prolonged period and/or at elevated temperatures for short-term exposure. The tensile and flexural properties of the aged composites were studied and indicated that an increase in the water content in a composite specimen leads to lots of rigidity or strength. There was a similar impact on the tensile and flexural properties of these composites reinforced with short natural fibres [41,42,43]. Scanning electron microscopy (SEM) analysis of different samples was also carried out which shows that debonding or the pull out of fibres was the reason for the failure of NFCs in commercial applications [41,43,44]. Considering the research findings presented in the literature, this present study focused on the fabrication of epoxy reinforced sisal fibre composites. The impact of humidity or moisture on epoxy reinforced sisal fibre composites under various environmental conditions has still not been addressed [45]. Moreover, different researchers used different hydrothermal ageing conditions, well-defined procedures are still not available, especially for sisal fibres. However, there is a limited number of studies that reported the impact of environmental variables on the mechanical characteristics of composite samples, and none of the research studies reported the simulation studies of NFC properties using ANSYS software [46]. Henceforth, an effort will be made in this study to define the hydrothermal characteristics for SFRE and explore the basic mechanisms behind the environmental ageing of SFRE composite and compared it with GFRE synthetic composite. Moreover, microstructural analysis of flexural tested GFRE and SFRE composite samples was carried out to identify the microstructural properties of the composites. In addition, the flexural properties of glass and sisal fibre reinforced epoxy composites under different hydrothermal conditions were analysed numerically using ANSYS software.

## 2. Materials and Methods

### 2.1. Fibre Reinforcement

Glass fibres and natural fibres were used as a reinforcement material for epoxy resin as shown in Figure 1. These composites were referred to as GFRE and SFRE. These fibre-reinforced epoxy resins were alkali resistant and possessed typical composition of silicon, boron, aluminium, and magnesium oxides. The natural fibres were chosen for reinforcement due to their low cost, high strength, and environmentally friendly nature. The sisal fibres needed chemical treatment using 6% sodium hydroxide (NaOH) before they could be incorporated into the epoxy resin to remove the oil and waxy substances and to improve fibre-matrix compatibility.

### 2.2. Epoxy Resin

In this study, low-cost epoxy resin (Kinetix R246TX, ATL composites Pty Ltd., Molendinar. QLD, Australia) and hardener (Kinetix H160, ATL composites Pty Ltd., Molendinar. QLD, Australia) were used. The resin has a low viscosity resin and is free of any residual solvent. Epoxy resin possesses aromatic groups, which makes it three times more powerful than polyvinyl resin. Moreover, it tends to withstand much greater impacts than either polyvinyl or polyester resin due to its high thermal stability. Hence, the epoxy resin had better mechanical properties than polyester or polyvinyl resins. Both fibres and the epoxy resin possessed effective adherence to each other, so it is proposed that the resultant composite will possess better mechanical characteristics than conventional glass fibre. The epoxy resin also exhibits better fiber-matrix interaction upon alkali treatment of reinforcing fibres. The hardener was used as a curing agent for curing epoxy at relatively low temperatures (room temperature). The widely used catalysts or hardeners are aromatic amines, aliphatic amines, and carboxylic anhydride. Both chemicals were provided by ATL composites Private Limited. As per the specifications, its pot life is nearly 2 h at room temperature and the demoulding time of almost 28 h. These properties make it very suitable to be used on a lab-scale and a commercial scale [47].

### 2.3. Preparation of Fibres

#### 2.3.1. Glass Fibres

Glass fibres were obtained as chopped sheet matting and they were used as received, without any chemical treatment. The fibres were sorted and cut into appropriate lengths for reinforcement, as shown in Figure 2.

#### 2.3.2. Sisal Fibres

The different steps involved in the preparation of sisal fibres are shown in Figure 3. The first step involved (a) unwinding, (b) sizing, and, (c) cutting of sisal fibre rope into various segments of 8 cm. The segments were unwound until the individual fibres were separated (d,e). These fibres had varying lengths so they were cut into small manageable pieces (f).

#### 2.3.3. Alkali Treatment

The final stage in preparing the sisal fibres was the alkali treatment using a 6% NaOH in a process referred to as mercerisation. Mercerisation enhances the surface roughness and increases the spread of reaction sites on the fibre as the cellulose in the fibre becomes exposed [47]. Raw sisal fibres are soaked in a 6% solution of caustic for 24 h and then washed several times with distilled water, the fibres are then dried in an oven at 40 °C for 24 h.

### 2.4. Composite Fabrication Steps

#### 2.4.1. Designing of Mould

The principal mould for composite fabrication was designed using SolidWorks^®^ software. Dimensional consistency is important for comparing the performance characteristics of different composites. The material used for the mould was ultrahigh molecular weight polyethene (UHMWPE) since it does not adhere to the epoxy and thus, fabricated composites can be separated easily from the mould. Moreover, this mould can be shaped using a machine with a high-speed water jet. The design information was sent to the QUT workshop, and the respective mould was received after fabrication. Figure 4 display the design parameters, mould dimensions, and respective parts.

#### 2.4.2. Composite Fabrication

The composite was fabricated using a UHMWPE mould having dimensions 80 mm × 10 mm (H × W) and a height of 4 mm as shown in Figure 5. The UHMWPE mould was first cleaned and then coated internally with a release agent. Sisal fibres were cut into 40 mm long lengths and were placed into the mould where the fibre volume ratio (Vf) was maintained at 30%. This is based on a previous study, which showed that the weight proportion of the fibre content is in the range of 23% to 34% influenced by several factors including the resin and fibre type and the manufacturing process [48]. In this study, the fibre ratio lies in between, and similar findings have been reported in other works [49]. The sisal fibres were aligned along the longest dimension of the mould (a). The epoxy resin was mixed with hardener and then poured into the mould (b). During this step, a steel spatula was used on the outer surface of the mould to remove any trapped air (c,d). The mould was then covered, and the resin was allowed to cure for 24 h at room temperature (e). The utmost effort were given to distribute fibers uniformly throughout the resin system. However, it is evident that fibers are not uniformly distributed and there are some voids were appeared in the cured samples. It should be noted here that the fiber distribution, number and size distribution of voids are the factors responsible that influence mechanical strength [19,50]. However, there are fewer voids or air bubbles in the current study and hence, have little influence on the overall strength of the composites. Moreover, during synthesis, voids often form during the alkali treatment and curing process [19,50].

### 2.5. Experimental Testing

#### 2.5.1. Hydrothermal Testing of GFRE and SFRE Samples

The hydrothermal testing of GFRE and SFRE samples was carried out by following the ASTM D5229 standard [51]. The composite samples were aged in a hydrothermal environment to evaluate their ability to absorb water, and to evaluate the effect of water on the respective mechanical properties. Table 3 shows the hydrothermal ageing scheme with respect to time and temperature.

The different stages of hydrothermal treatment are shown in Figure 6. Before the hydrothermal ageing tests, free water was removed from all of the samples by conditioning in an oven at around 100 °C for 24 h as shown Figure 6a. Then the samples of GFRE and SFRE were weighed and immersed in distilled water for two different periods as depicted in Figure 6b,c. Thereafter, the conditioned samples were kept in an oven at 60 °C (Figure 6d). After the soaking time, they were dried and weighed again for the second time as shown in Figure 7e. The GFRE and SFRE samples were tested under four hydrothermal conditions and listed in Table 4.

#### 2.5.2. Microstructure Analysis of GFRE and SFRE Samples

The microstructures of the GFRE and SFRE composite samples was analyzed using a Digitech^®^ 5MP USB Microscope Camera, Sydney, Australia, as shown in Figure 7. It is easy to operate and can be zoomed in to analyze surface features. This technique provides information about the surface morphology, fibre pull-out, and fracture patterns of the tested samples. 

#### 2.5.3. Mechanical Testing of GFRE and SFRE Samples

Three-point bending tests were carried out according to ASTM D790-07 [52] to investigate the mechanical characteristics of the fabricated GFRE and SFRE composites by using an Instron© 5566 material testing machine (Ithaca, NY 14850, USA). A schematic layout and photograph of the flexural testing setup are shown in Figure 8 and Figure 9, respectively. 

The machine applies the force onto the center of the specimen as shown in Figure 9 and Figure 10 and the resultant applied force in the sample is estimated by the machine which is then presented in terms of flexural stress, flexural strain, and flexural modulus. The flexural testing was carried out with a deformation rate of 2 mm/min at a distance of 15 mm from both ends. The maximum flexural stress, flexural strain, and flexural modulus at the midpoint of the samples were calculated by using the following equations:σ*_f_* = 3PL/2bd^2^(1)
ε*_f_* = 6Dd/L^2^(2)
E*_f_* = mL^3^/4bd^3^(3)
where,

σ*_f_* = Flexural stress (MPa)

ε*_f_* = Flexural strain (mm/mm)

E*_f_* = flextural modulus of elasticity

P = maximum load at a given point (N)

L = distance b/w supports (mm)

b = width of tested beam (mm)

d = thickness of tested beam (mm)

D = measured deflection (mm)

m = slope (N/mm)

Multiple flexural tests were carried out using GFRE samples, as shown in Figure 10. Similarly, SFRE was also tested as shown in Figure 11. The data were collected and the mean values were taken for further analysis.


Figure 10The flexural testing of GFRE samples.
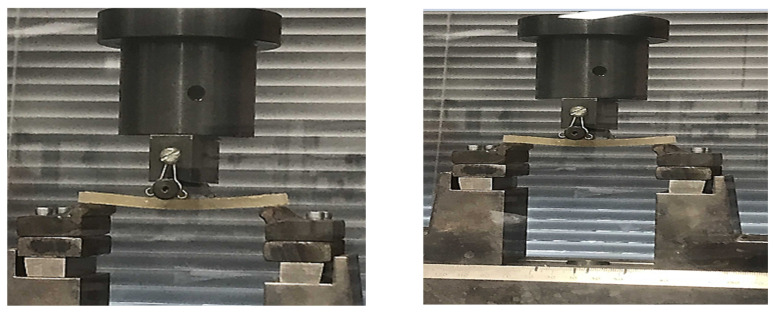

Figure 11The flexural testing of SFRE samples.
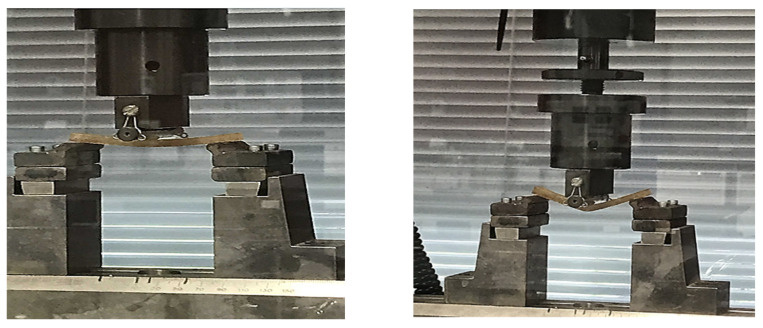



### 2.6. Numerical Simulation in ANSYS

The results obtained by hydrothermal ageing of fibre composites were compared with the numerical studies conducted in modern finite element ANSYS software [46], as shown in Figure 12. A numerical analysis of the hydrothermal characteristics was carried out using ANSYS software. ANSYS hydrothermal analysis module fully integrated into the ANSYS Workbench platform. ANSYS is one of the most powerful software in the hydrothermal analysis field that is integrated with hydrothermal performance parameters of more than 200 metal and composite materials, which is very useful. The numerical analysis steps can be divided into these parameters: (a) creating finite element models; (b) importing material properties; (c) defining parameters for performance and analysis; (d) running hydrothermal analysis; (e) exporting the numerical results; and (f) hydrothermal result evaluation. The element type of the numerical models was SOLID185 which has plasticity, hyper-elasticity, stress stiffening, creep, large deflection, and large strain capabilities [46]. The mesh size and element number have been chosen after the mesh convergence analysis. Table 5 shows the material properties of each material utilized during the simulation.

## 3. Results and Discussion

### 3.1. Effect of Water Absorption

A water absorption test was carried out on the GFRE and SFRE. The samples were kept in water for 24 h at room temperature. The results revealed that SFRE absorbed more moisture than GFRE. The moisture absorbed was as high as 4%, as shown in Figure 13. Sen and Reddy [53] and Azwa, Yousif [54] reported similar behaviour in their research articles relating to natural fibre composites (NFCs), where NFCs were absorbed more moisture compared to the glass fibre reinforced composites (GFRCs), and the absorptions were varied between 0.7 to 2% for 24 h of soaking. In this study, the SFRE absorbed slightly more moisture as compared to the previous studies. This is due to the presence of hemicellulose in sisal fibres and the hollow cavities in the natural fibres.

### 3.2. Flexural Behavior and Durability of GFRE under Different Hydrothermal Conditions

The flexural behaviour of GFRE samples under hydrothermal conditions is presented as a stress vs. deflection curve in Figure 14, where three samples were tested at each hydrothermal condition. All GFRE samples showed a linear relationship for specific values of deflection at respective loads. This behaviour continued up to the elastic limit, after which it reached the plastic region, leading to the point of ultimate strength or fracture. In all of these observations, the minimum to a maximum range of ultimate strength was 160–205 MPa, and the deflection values were between ~35 to ~13 mm. For each case, all of the samples were fabricated using the same procedure and tested under the same hydrothermal conditions. However, some samples showed variable trends due to different slopes under C2, each trend had an elastic region, followed by plastic deformation and fracture. 

### 3.3. Flexural Behavior and Durability of SFRE under Different Hydrothermal Conditions

The stress vs. deflection curve shown in Figure 15 exhibited the flexure behaviour of SFRE, just as in GFRE. The three samples of untreated SFRE exhibited a linear relationship for most loads, followed by a sharp decline at the breakage point. All of the samples had a similar rate of deflection that was directly proportional to the respective loads. This behaviour was exhibited to the limit of proportionality, leading to the elastic limit. Beyond this limit, a plastic zone was formed, and samples began to yield beyond their ultimate flexural strength. The average ultimate flexural strength was ~107 MPa. The behaviour of the samples tested under hydrothermal C2–C4, as shown in Figure 15b,d, were similar but the average ultimate flexural strength decreased by ~30%. The graphical trends of the samples also were similar, but the samples treated under C1 were different as the plastic deformation region was more widespread and there was a range of yield points. This difference was attributed to the absorption of moisture during hydrothermal ageing.

### 3.4. Flexural Behavior and Durability Comparison between the GFRE and SFRE

The flexural strength of the GFRE samples treated under four different hydrothermal conditions is compared in Figure 16a. The bar chart showed that the untreated sample exhibited a maximum flexural strength of ~159 MPa, but with the hydrothermal treatment, the flexural strengths decreased. The sample with C4 had a minimum flexural strength of ~91 MPa. There was a reduction of 42% in the flexural strength GFRE samples as the hydrothermal treatment continued from C1 to C4. The flexural strengths of the four SFRE samples are compared in Figure 15b and show an overall improvement in the flexural strength of the epoxy resin. Cao, Shibata [55] and Sanjay, Arpitha [36] reported similar findings. The maximum flexural strength of the unconditioned sample was ~107 MPa, whereas the sample treated in C4 has a maximum flexural strength of ~61 MPa. Overall, the flexural strength of the GFRE samples was higher due to better fibre distribution and good matrix-fibre interaction [10]. Hydrothermal ageing reduced the flexural strength of the composites by 43%, which was similar to the GFRE samples. On this basis, it could be inferred that the durability of the GFRE and SFRE samples was equally affected by the hydrothermal treatment. 

It should be noted that, unlike the GFRE, hydrothermal ageing had no real effect on the SFRE samples up to C3, and the flexural strength only decreased by 7%. However, there was a large change under C4 and a sharp decline of 37% in flexural strength; the SFRE samples can perform well for 40 days under 100% humidity at a temperature of 60 °C. The variations in flexural strength can be explained based on the moisture uptake exhibited by natural fibres. Upon exposure to moisture, water penetrates by the capillary mechanism and forms hydrogen bonding with the hydrophilic groups of natural fibres. The mechanism of water uptake by sisal fibres in the epoxy composite presented elsewhere [42]. This process is catalyzed at elevated temperatures. This results in a weak interfacial adhesion between fibres and matrix, which marks the onset of composite degradation. The cellulose fibres then begin to swell which leads to the development of micro-cracks around the swollen fibres. These micro-cracks facilitate further water transportation and leaching of water-soluble substances from fibres which ultimately causes the matrix and fibres to de-bond [56,57]. Researchers working on other natural fibres reported similar observations [39,57].

### 3.5. Comparison between Experimental and Numerical Results

The flexural strengths of GFRE and SFRE under various hydrothermal conditions were modelled numerically using ANSYS software as mentioned before. The analysed models in the software are shown in Figure 17. A comparison of the experimental and numerical flexural strengths of GFRE obtained is shown in Figure 18. Both values showed a similarity in behaviour across the conditions C1 to C3. In the numerical analysis, there was a slight decrease in flexural strength between the first two conditions (C1–C2). This behaviour continued with a substantial decrease (77%) under C3. The experimental flexural strength decreased similarly, but it was higher than the numerical value under C3.

A comparison of the experimental and numerical flexural strengths of the SFRE obtained in this study is also shown in Figure 19. There was a slight agreement between these two results with similar behaviour from C1 to C2. In the numerical analysis, there was a decrease of only 15% in the flexural strength from C1 to C2 and a substantial decrease of 84% under C3. In the experimental studies, however, the flexural strength fluctuated slightly but remained almost the same.

### 3.6. Microstructure Analysis of Flexural Tested Samples

The microstructural images of the GFRE and SFRE composite samples obtained using the Digitech^®^ 5MP USB Microscope Camera, Australia are shown in Figure 20 and Figure 21. In the GFRE, the glass fibres were more refined and evenly distributed through the epoxy matrix. The coarse sisal fibres were also randomly distributed, but there were also agglomerates of sisal fibres in the samples. There was also a formation of bubbles due to alkali treatment (despite both figures showing that the fibre pull-out mechanism was mainly responsible for flexural breaking, especially with the sisal fibres). This observation endorsed the moisture absorption and de-bonding characteristics of SFRE and GFRE discussed in the flexural testing section previously.

### 3.7. SEM Analysis

Figure 22 (SEM images) shows the sisal fibre before and after treatment. Note that the alkali treatment has enhanced the surface roughness by removing the layer of wax and other impurities.

## 4. Conclusions

This study presents a comprehensive study on the hydrothermal ageing and mechanical characteristics of SFRE composites and then compared them with the hydrothermal ageing and mechanical characteristics of GFRE composites. In addition, a first-of-its-kind numerical model has been developed to study the hydrothermal ageing and mechanical characteristics of SFRE, along with GFRE, through modelling and simulating with ANSYS software. Moreover, the microstructural properties of the composites were identified by microstructural analysis of flexural tested GFRE and SFRE composites. Based on the results of this study, the following conclusions can be drawn: Both experimental and numerical results revealed that the flexural properties of both types of GFRE and SFRE composites are affected quite significantly by short- or long-term hydrothermal treatment.GFRE experienced moisture uptake and fibre-matrix de-bonding, but not as much as SFRE composites. This means the amount of fibre is important since it determines the final mechanical characteristics of the composite.SFRE samples did not experience very much hydrothermal ageing up to C3 at a temperature of 60 °C and humidity of 100% during 1000 h; their experimental and numerical flexural strengths were comparable to GFRE composites.This study showed that SFRE exhibited moisture uptake and fibre-matrix de-bonding, as do GFRE, but with less severity, as their flexural properties are affected by short or long-term hydrothermal treatment. In addition, applications involving flexural loads in natural environments were found to be more reliable and durable with glass fibre-based composites.Microstructure analysis revealed the formation of bubbles and fibre pull-out mechanism in both GFRE and SFRE composites, especially with the SFRE, which were mainly responsible for the moisture absorption and de-bonding under flexural loading characteristics of GFRE and SFRE composites.

## Figures and Tables

**Figure 1 polymers-14-02024-f001:**
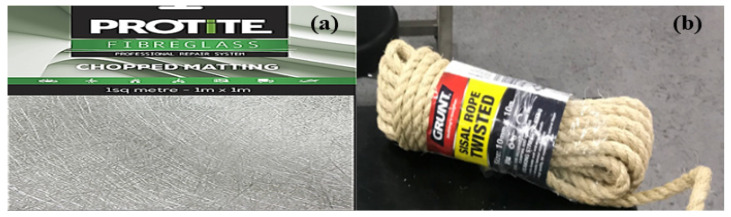
(**a**) Glass fibres rope as a reinforcement for epoxy and (**b**) sisal fibres rope as a reinforcement for epoxy.

**Figure 2 polymers-14-02024-f002:**
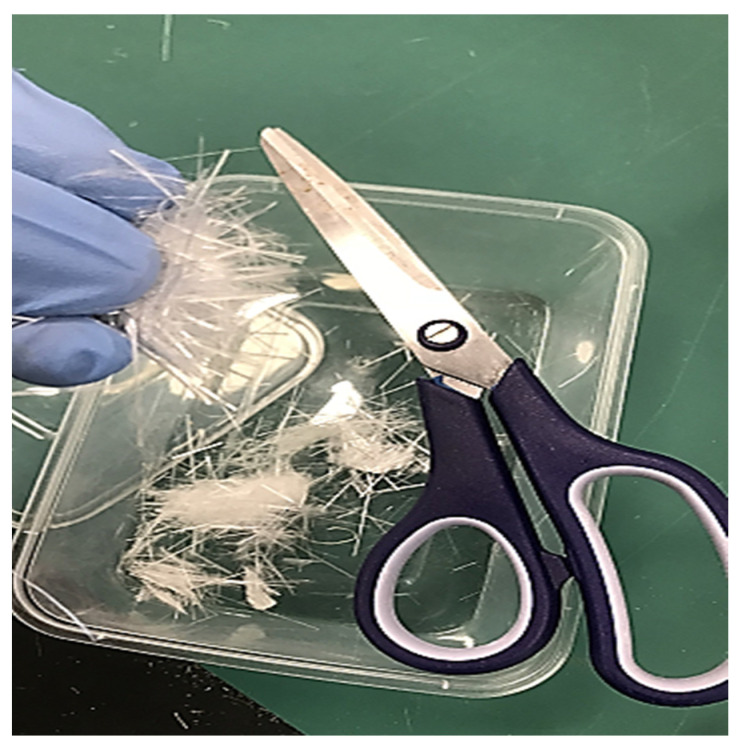
Sorting and cutting of glass fibres.

**Figure 3 polymers-14-02024-f003:**
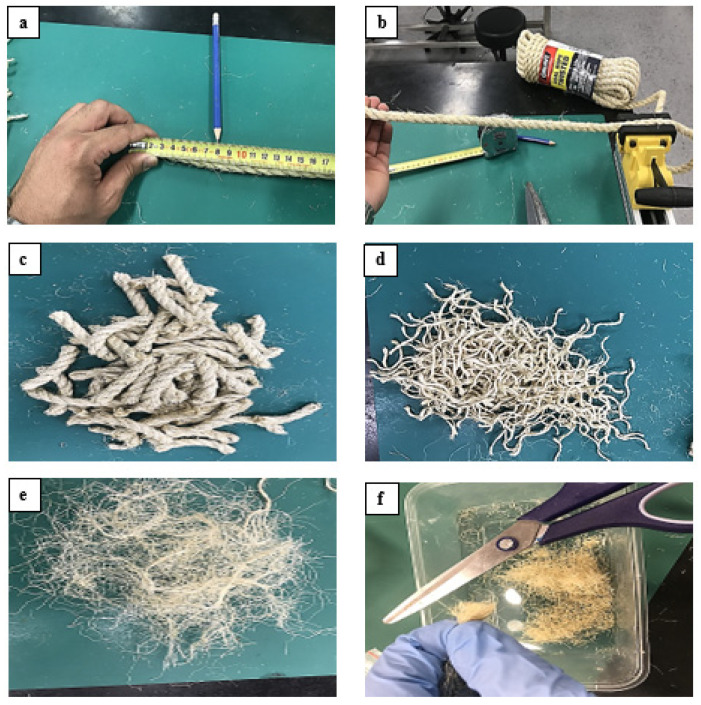
Different steps in the preparation of sisal fibres (**a**) unwinding, (**b**) sizing, and, (**c**) cutting (**d**,**e**) separation and (**f**) small manageable pieces.

**Figure 4 polymers-14-02024-f004:**
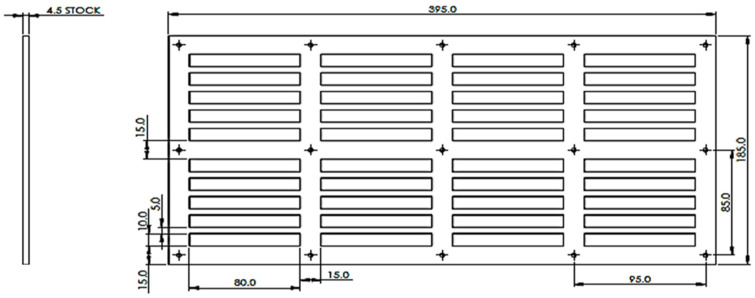
Images of a mould designed using SolidWorks^®^ software.

**Figure 5 polymers-14-02024-f005:**
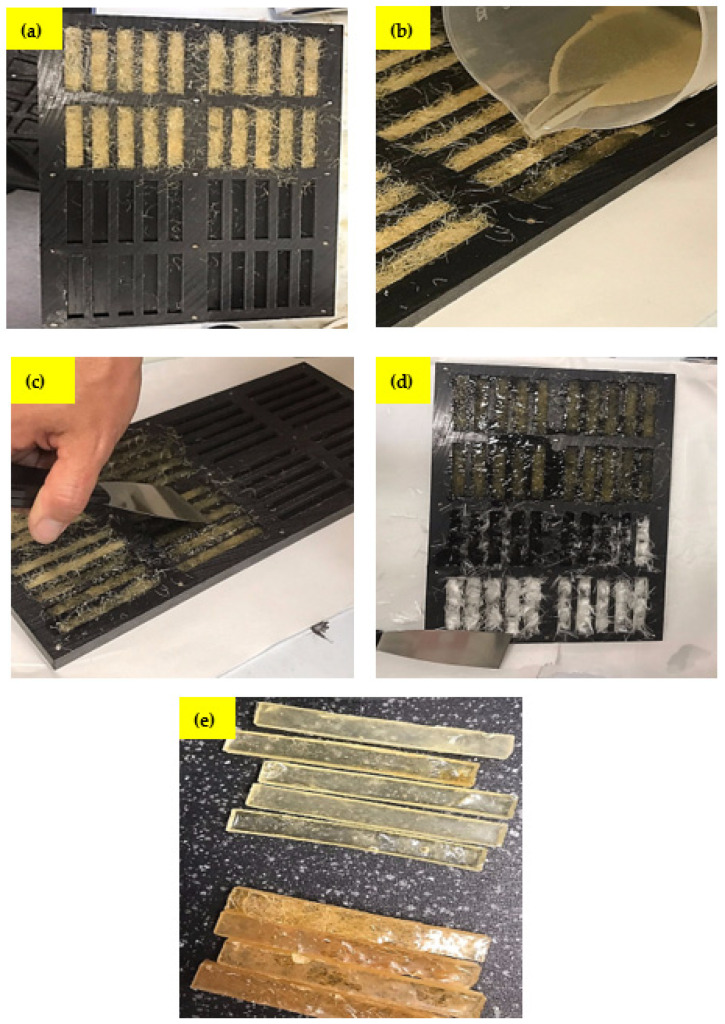
Different stages of composite fabrication (**a**) fibre aligned in the mould (**b**) epoxy with hardener pour into the mould (**c**,**d**) trap air removal (**e**) cured samples.

**Figure 6 polymers-14-02024-f006:**
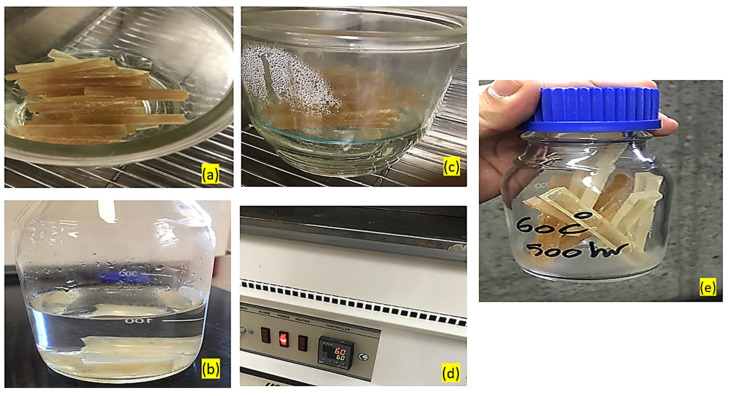
Different stages of hydrothermal testing (**a**) samples in the oven at 100 °C for 24 h, (**b**,**c**) samples immersed in distilled water (**d**) oven at 60 °C and (**e**) soaking time.

**Figure 7 polymers-14-02024-f007:**
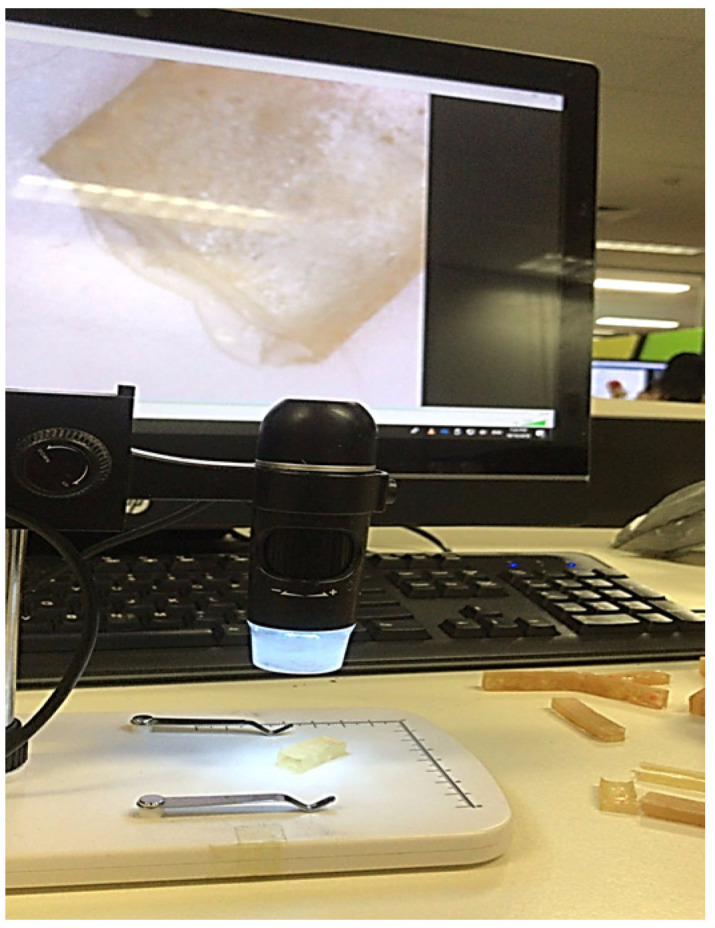
Microstructure Analysis (A Digitech^®^ 5MP USB Microscope Camera).

**Figure 8 polymers-14-02024-f008:**
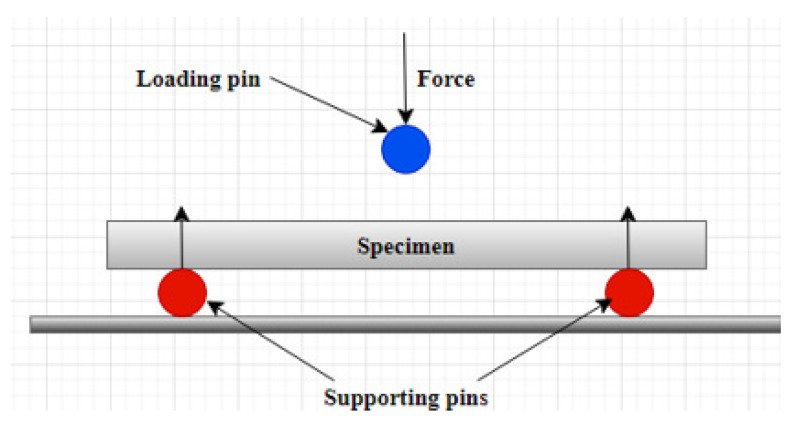
The schematic diagram of three-point flexure testing using a sample with dimensions 80 mm× 10 mm× 4 mm (L × W × H).

**Figure 9 polymers-14-02024-f009:**
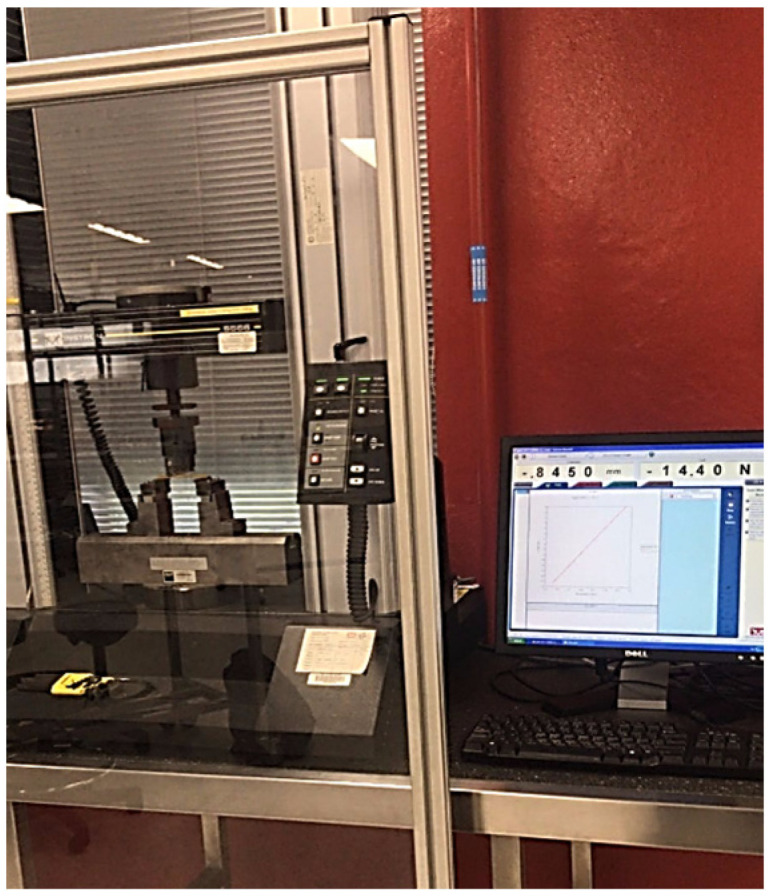
Photograph of the flexural test by using the Instron© 5566.

**Figure 12 polymers-14-02024-f012:**
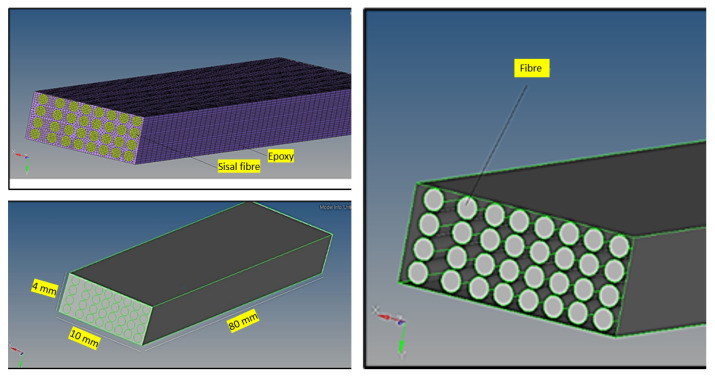
Different stages of model development in ANSYS.

**Figure 13 polymers-14-02024-f013:**
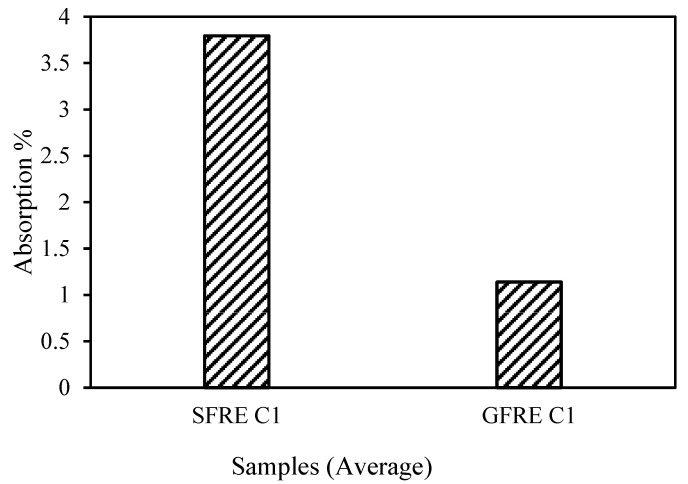
Preliminary study of the capability of sisal and glass fibres to absorb moisture at C1.

**Figure 14 polymers-14-02024-f014:**
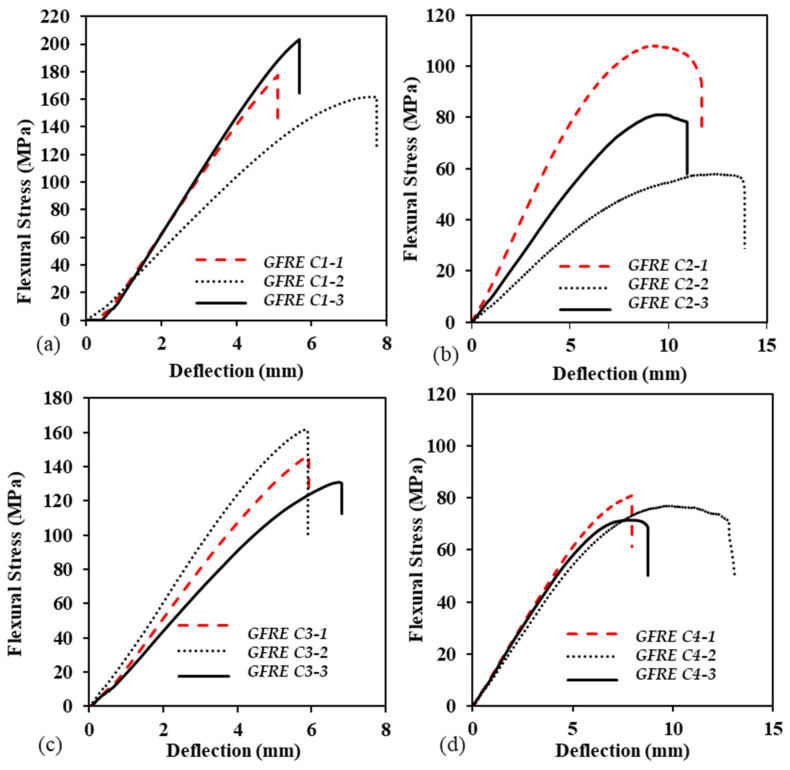
Flexural behaviour of GFRE under (**a**) C1, (**b**) C2, (**c**) C3, and (**d**) C4.

**Figure 15 polymers-14-02024-f015:**
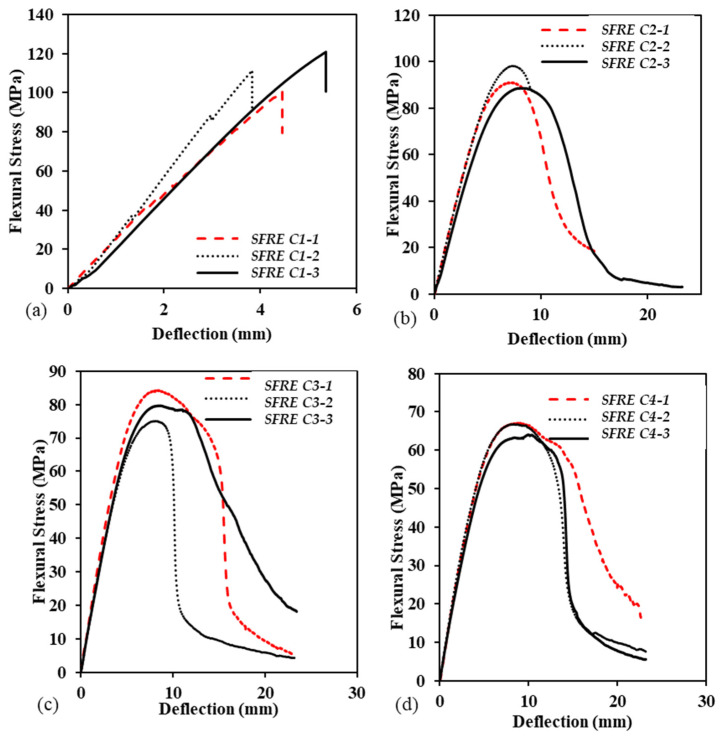
Flexural behaviour of SFRE samples under various hydrothermal conditions, (**a**) C1, (**b**) C2, (**c**) C3, (**d**) C4.

**Figure 16 polymers-14-02024-f016:**
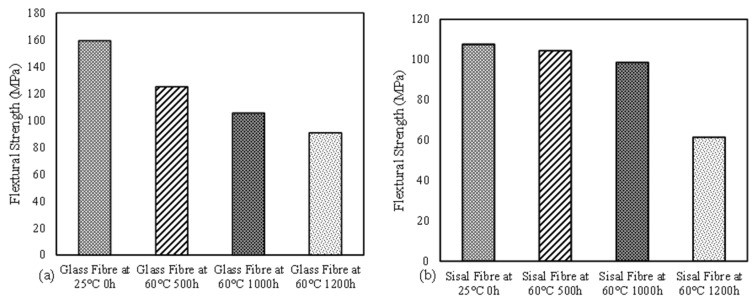
Flexural strengths of GFRE (**a**) and SFRE (**b**) samples under hydrothermal testing.

**Figure 17 polymers-14-02024-f017:**
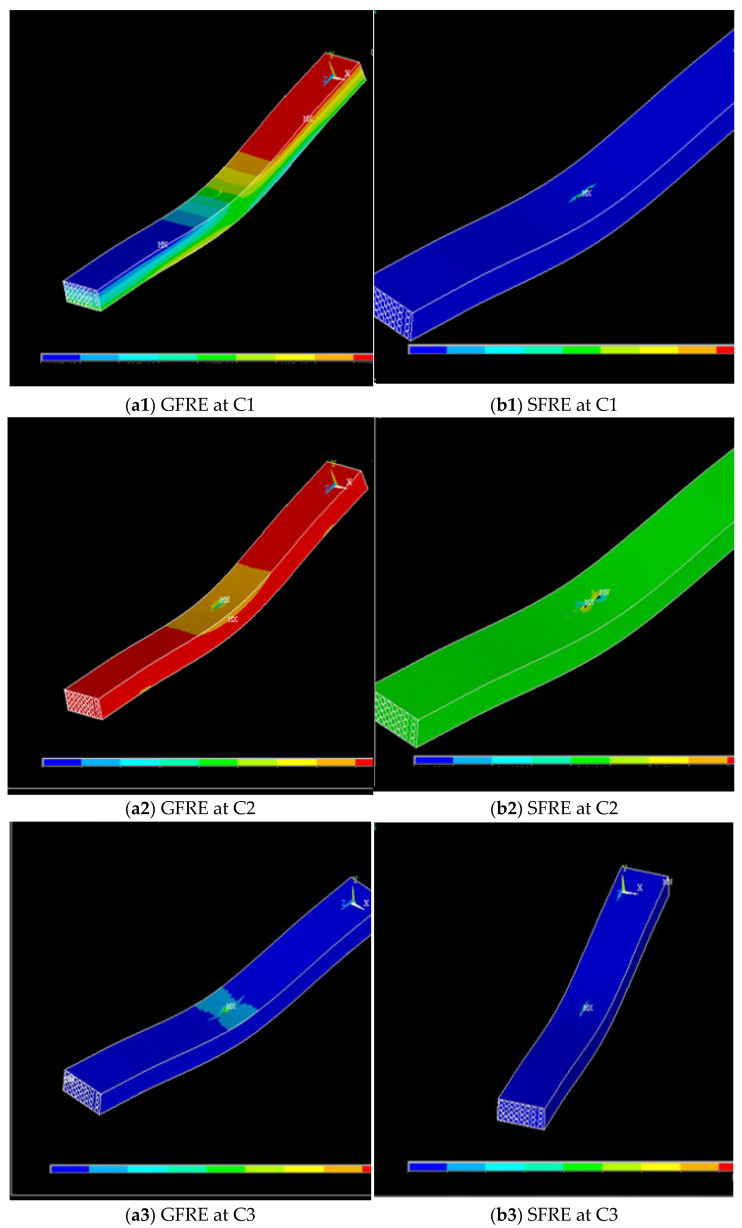
(**a1**–**a4**) Stress distribution of GFRE at different conditions and (**b1**–**b4**) stress distribution of SFRE at different conditions.

**Figure 18 polymers-14-02024-f018:**
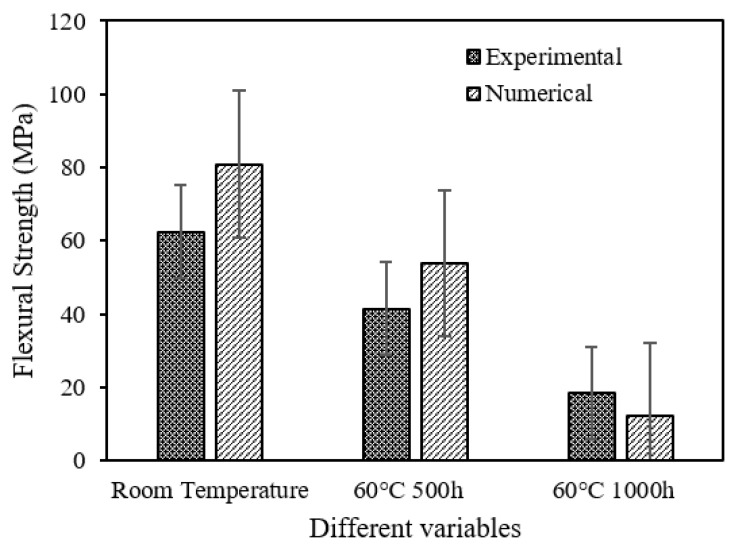
Comparison between the experimental and numerical flexural strength of GFRE.

**Figure 19 polymers-14-02024-f019:**
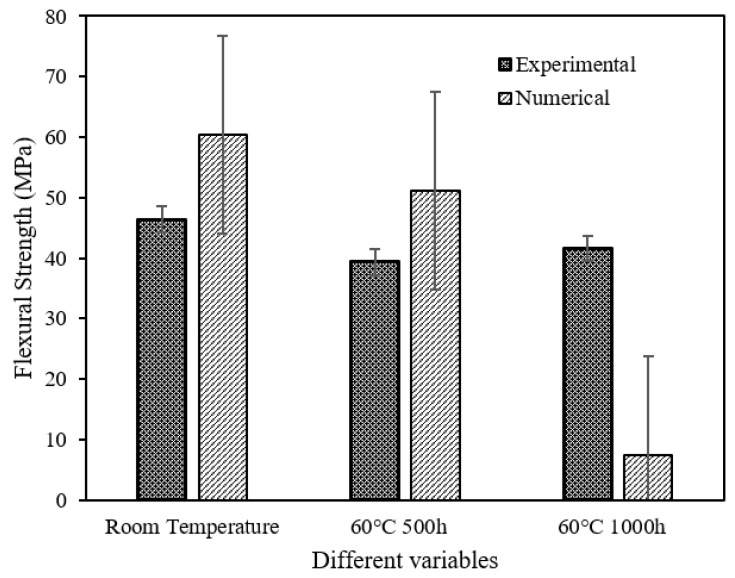
Comparison between experimental and numerical flexural strength of SFRE.

**Figure 20 polymers-14-02024-f020:**
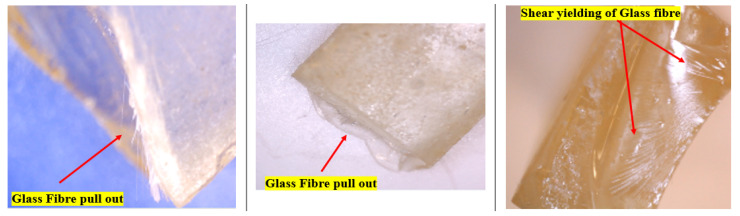
Microstructural images of the GFRE composite samples.

**Figure 21 polymers-14-02024-f021:**
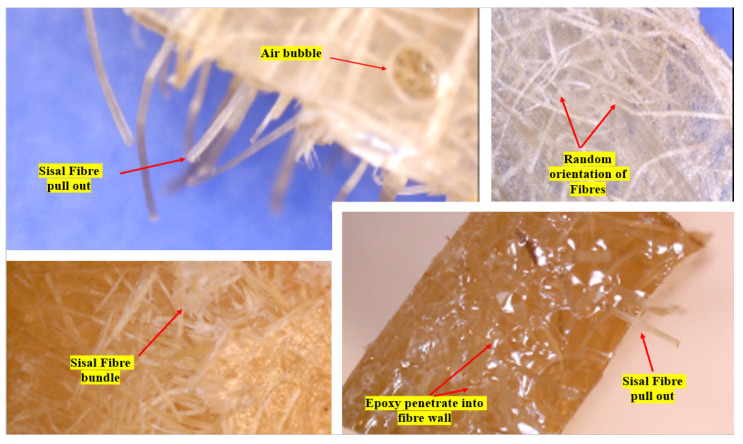
Microstructural images of the SFRE composite samples.

**Figure 22 polymers-14-02024-f022:**
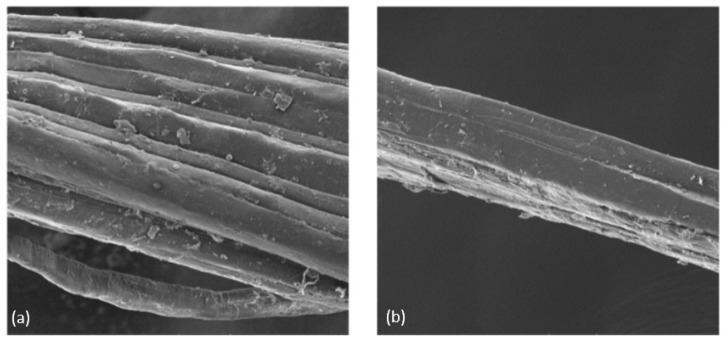
(**a**) Untreated fibre and (**b**) treated fibre.

**Table 1 polymers-14-02024-t001:** Mechanical and physical properties of sisal fibre [25,33].

SI. No.	Chemical Constituents	Sisal Fibre (%)
1	Density (g/cm^3^)	1.34–1.5
2	Fiber diameter (µm)	145–440
3	Elongation (%)	1.54–3.85
4	Young’s modulus (GPa)	9.6–38
5	Tensile strength (MPa)	400–700

**Table 2 polymers-14-02024-t002:** Chemical composition of sisal fibres [34].

SI. No.	Chemical Constituents	Sisal Fibre (%)
1	Cellulose (%)	41.6–62.6
2	Lignin (%)	11.4–14.6
3	Hemi cellulose (%)	9.2–14.6

**Table 3 polymers-14-02024-t003:** Hydrothermal ageing scheme of GFRE and SFRE with respect to time and temperature.

Time (h)	No. of Samples		Temperature (°C)
	GFRE Samples	SFRE Samples	
0	3	3	25
500	3	3	60
1000	3	3	60
1200	3	3	60

**Table 4 polymers-14-02024-t004:** Hydrothermal Conditions.

Condition Number	Parameters
Condition (C1)	Temperature: 25 °C
	Time Duration: 0 h
	Relative Humidity: 0%
Condition (C2)	Temperature: 60 °C
	Time Duration: 500 h
	Relative Humidity: 100%
Condition (C3)	Temperature: 60 °C
	Time Duration: 1000 h
	Relative Humidity: 100%
Condition (C4)	Temperature: 60 °C
	Time Duration: 1200 h
	Relative Humidity: 100%

**Table 5 polymers-14-02024-t005:** Property materials utilized in ANSYS [46].

Materials	Density (tonne/mm^3^)	Young’s Modulus (GPa)	Poisson Ratio	Thermal Conductivity (W/mk)
Glass Fibre	2.5 × 10^−9^	72	0.21	1.2
Sisal Fibre	1.45 × 10^−9^	38	0.286	0.25
Epoxy	1.25 × 10^−9^	3.5	0.0197	0.35

## Data Availability

Not applicable.

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
