# Peer review of "An Experimental and Numerical Investigation into the Durability of Fibre/Polymer Composites with Synthetic and Natural Fibres"

_polymers, 2022, doi:10.3390/polym14102024_

Round 1
Reviewer 1 Report
The authors should describe the properties of the fibers. Are the properties same as virgin fibers? As they are taken from twisted yarns/ropes, are there any loss of mechanical properties?
Such limitations should be clearly indicated. Though such composites are fine for experiments, they do not reflect the exact performance of bulk fibers. The fibers of sisal must have been chemically treated before the ropes were made. Since the authors bought them from market, they may not be aware of those treatments and related loss of mechanicalproperties.
As far as the ANSYS model in Fig. 13 is concerned, how do the authors assume such uniform fiber arrangement? How can all the fibers be of same dimensions?
The figure numbering has gone wrong after Fig. 19.
In spite of all these observations, the paper is interesting and should be allowed for a major revision.
Author Response
The authors would like to thank the reviewer for the efforts and time in reviewing the paper. In the attached document, the authors have attempted to answer the reviewers’ comments.

Reviewer 2 Report
- Literature review on natural fiber composites should be enriched.
- Add a table showing mechanical and physical properties of sisal fibers
- What was fiber volume fraction in composite samples?
- From the figure 6, it is evident that fibers are not uniformly distributed and there are voids as well. Please comments on it.
- Figure 2, the error in 60°C 1000hr is very high, reason?
- English needs a revision
Author Response

(The authors gave the same response as above.)

Round 2
Reviewer 1 Report
Can be accepted.